# Dengue transmission heterogeneity across Indonesia's archipelago: Climate-driven spatiotemporal patterns and policy implications

**Bimandra A. Djaafara**[1,2]\*, **Iqbal R.F. Elyazar**[2,3], **Fadjar S.M. Silalahi**[4], **Asik Surya**[5], **Agus Handito**[4], **Burhannudin Thohir**[4], **Desfalina Aryani**[4], **Mushtofa Kamal**[6], **Aditya L. Ramadona**[7], **Dyana Gunawan**[4], **Hipokrates**[4], **Anzala Khoirun Nisa**[4], **Edi Prianto**[4], **Iriani Samad**[4], **Agus Sugiarto**[4], **Kimberly Fornace**[1], **Hannah E. Clapham**[1], **Nuno R. Faria**[8], **Swapnil Mishra**[1]\*

**1** Saw Swee Hock School of Public Health, National University of Singapore and National University Health System, Singapore, Singapore, **2** Oxford University Clinical Research Unit Indonesia, Faculty of Medicine, University of Indonesia, Jakarta, Indonesia, **3** Monash University Indonesia, Tangerang, Banten, Indonesia, **4** Arbovirus Working Team, Ministry of Health of the Republic of Indonesia, Jakarta, Indonesia, **5** Joint Coalition Against Dengue (KOBAR Lawan Dengue), Jakarta, Indonesia, **6** Public Health Emergency Operation Center, Ministry of Health of the Republic of Indonesia, Jakarta, Indonesia, **7** Department of Health Behavior, Environment and Social Medicine, Faculty of Medicine, Public Health and Nursing, Universitas Gadjah Mada, Yogyakarta, Indonesia, **8** MRC Centre for Global Infectious Disease Analysis, School of Public Health, Imperial College London, London, United Kingdom

\* bimandra.djaafara@nus.edu.sg (BAD); swapnil.mishra@nus.edu.sg (SM)

## Abstract

Indonesia has the highest dengue burden in Southeast Asia, with 488 of 514 districts reporting cases annually across its 17,000-island archipelago. Despite this substantial burden, spatiotemporal transmission patterns remain poorly characterised. We analysed province-level dengue surveillance data (2010–2024) from Indonesia's Ministry of Health alongside local and regional climate variables to characterise heterogeneity in dengue periodicity and identify provinces where climate-based early warning may be feasible. Using wavelet phase analysis, dynamic time warping clustering, and distributed lag non-linear models, we examined relationships between climate and dengue incidence across 34 provinces. A systematic west-to-east gradient in dengue wave timing was identified, with Northern Sumatran provinces peaking earlier than other provinces, aligning with Australian-Asian monsoon progression. This gradient was robust in western Indonesia (Spearman $\rho = 0.7$ between longitude and phase lag) but weakened in eastern provinces. Multi-annual outbreak peaks (2015–2016, 2023–2024) coincided with strong El Niño events, with mean incidence during strong El Niño years was 96% higher than other years. The Indian Ocean Dipole showed no significant association. Phase coherence analysis identified 18 provinces where precipitation-dengue timing was sufficiently consistent (coherence ≥0.85) for potential early warning applications and DLNM confirmed significant dose-response associations in 11 of these. Indonesia's dengue-climate relationships exhibit structured heterogeneity that precludes uniform national prediction approaches but may

**Data availability statement:** Data availability Province-level monthly dengue incidence data up to October 2024 are available at Open Dengue (https://opendengue.org/). Climate data were obtained from ERA5-Land (Copernicus Climate Data Store, https://cds.climate.copernicus.eu/). Detailed dengue surveillance data beyond what is available through Open Dengue remain the property of the Ministry of Health Indonesia and are subject to data sharing restrictions. Researchers interested in accessing additional surveillance data may contact the Arbovirus working team of the Ministry of Health at arbovirosis.subdit@gmail.com. Code availability The code used for data analysis in this study is available at https://github.com/mlgh-sg/dengue-climate-indonesia.

**Funding:** BAD is supported by funding from the Program for Research in Epidemic Preparedness and Response (PREPARE) from the Ministry of Health, Singapore (A-8000642-01-00 PREPARE S2-2024-002). NRF and SM was supported by a World Health Organization–Temasek Foundation Collaboration Framework Grant. NRF was supported by a Wellcome Trust Dengue and Zika Immunology and Genomics Multi-Country Network (DeZi Network) (316633/Z/24/Z). SM acknowledges support from the National Research Foundation, Singapore, under its NRF FELLOWSHIP (NRF-NRFF15-2023-0010) awarded to him. IRFE is supported by the Wellcome Africa Asia Program Vietnam (106680/Z/I4/Z) and the Strategic Partnership for Prevention, Surveillance and Response to Infectious Diseases across the Indo-Pacific Region (SPARKLE). We acknowledge additional funding support from the NUS Climate, Environment and Health programme. The funders had no role in study design, data collection and analysis, decision to publish, or preparation of the manuscript.

**Competing interests:** The authors have declared that no competing interests exist.

enable province-specific early warning in high-coherence areas. A two-tier system combining ENSO monitoring for strategic preparedness with local climate monitoring for tactical intervention timing could improve outbreak response across Indonesia's diverse epidemiological landscapes.

## Author summary

Indonesia reports hundreds of thousands of dengue cases annually, ranking among the highest-burden countries in Southeast Asia. Effective disease control requires understanding when and where outbreaks occur, yet Indonesia's vast archipelago spanning over 5,000 kilometres presents unique challenges for prediction and response. We analysed 15 years of dengue surveillance data across all 34 Indonesian provinces to characterise how outbreak timing varies geographically and relates to climate patterns. We found a consistent west-to-east gradient in outbreak timing across western Indonesia, with provinces in Sumatra peaking up to four months earlier than those in Java and Bali, following the progression of the monsoon system. However, this pattern breaks down in eastern Indonesia, where climate regimes differ markedly. El Niño events were strongly associated with nationwide outbreak years, with incidence nearly doubling during strong El Niño periods. We identified 18 provinces where rainfall-dengue timing relationships remained consistent across years, suggesting where climate-based early warning systems may be most feasible. Our findings support province-specific rather than uniform national approaches to dengue preparedness, with coordination between early- and late-peaking provinces to optimise resource deployment.

## Background

Dengue represents one of the most rapidly expanding vector-borne diseases globally, with Indonesia becoming the country with the highest dengue burden in Southeast Asia [1]. As an archipelagic nation spanning over 17,000 islands across 5,000 kilometres of longitude, mostly positioned along the equator, Indonesia presents unique epidemiological and public health challenges not found in continental countries [2,3]. The country's vast geographic diversity, ranging from dense urban centres to remote island communities and from coastal to mountainous regions, creates heterogeneous landscapes where dengue transmission remains poorly understood.

Indonesia's dengue burden is substantial and continues to grow. A multi-centre cohort study conducted across hospitals in Java and Sulawesi found that 32% of hospitalised patients with acute fever tested positive for dengue [4]. All four dengue serotypes were found to be circulating across the country, although serotype dominance varies between regions, highlighting the hyperendemic nature of transmission in Indonesia [4,5]. Currently, 488 of Indonesia's 514 districts in 38 provinces report dengue cases annually [6], demonstrating the disease's widespread presence across

the archipelago. The large 2024 dengue outbreak in Indonesia serves as a stark reminder of the country's vulnerability to this disease, with approximately 250,000 confirmed cases (a 2.2-fold increase from 2023) and over 1,400 deaths reported nationally [7,8].

Despite the substantial burden of dengue in Indonesia, the spatial and temporal patterns of transmission across the archipelago remain poorly characterised. While the spatial heterogeneity of dengue has been recognised [9], the specific drivers causing different transmission patterns across the country's diverse geography have not been systematically studied. Previous research on spatiotemporal dengue patterns has focused on other regions, including comprehensive analyses in Brazil [10], Costa Rica [11], and broader Southeast Asia [12] that excluded Indonesia from their analyses. This knowledge gap represents a critical limitation for public health planning in a country where dengue poses such a significant threat.

Understanding dengue periodicity and transmission patterns is essential for effective outbreak preparedness and resource allocation [13]. In a country as geographically and climatically heterogeneous as Indonesia, region-specific transmission dynamics likely require tailored prevention and control strategies. The timing of dengue seasons and the influence of climate variables may vary substantially across Indonesia's diverse provinces, necessitating province-level analysis to better inform targeted interventions.

In this study, we characterised the periodicity and spatiotemporal transmission patterns of dengue across Indonesia's provinces by analysing fifteen years of surveillance data (2010–2024) alongside local and regional climate variables. Our objectives were to: 1) characterise the spatial and temporal heterogeneity in dengue periodicity across Indonesia's provinces, 2) examine relationships between local and regional climate variables and dengue transmission patterns, and 3) identify provinces where climate-dengue timing relationships are sufficiently consistent to support early warning applications.

## Methods

### Study area and administrative boundaries

Indonesia comprises 34 provinces under pre-2022 administrative boundaries (map of geographical regions are shown in **Fig A** in S1 Appendix). We excluded four provinces established in 2022 from the reorganisation of Papua (South Papua, Central Papua, Highland Papua, and Southwest Papua) due to substantial missing data. For this study, the Papua region includes Papua and West Papua provinces.

### Epidemiological data

We obtained monthly dengue incidence data for 2010–2024 (180 time points) from the Arbovirus Working Team of the Indonesian Ministry of Health and the Open Dengue platform (except final months of 2024) [14]. The data included clinically diagnosed cases of dengue haemorrhagic fever (DHF) reported through Indonesia's national dengue surveillance system [15,16]. Laboratory confirmation is uncommon, with diagnosis typically based on clinical presentation and basic haematology, supplemented by rapid diagnostic tests where available. We calculated incidence rates per 100,000 population using linear interpolation and extrapolation of 2010 and 2020 census data by age group and sex [17,18].

### Climate data

**Local climate variables.** We used ERA5-Land reanalysis data from the European Centre for Medium-Range Weather Forecasts (ECMWF) [19], aggregated monthly. Monthly precipitation represents total precipitation (mm). Monthly temperature represents the average daily mean 2-metre temperature (°C). Monthly relative humidity represents the average daily mean relative humidity (%). Relative humidity was calculated from 2-metre temperature (T) and dewpoint temperature (Td) using the formula [20]:

$$RH = 100 \times exp(\frac{17.625 \times T_d}{243.04 + T_d})/exp(\frac{17.625 \times T}{243.04 + T})$$

Climate variables were aggregated to the province level using population-weighted averages from the LandScan Global Population Database [21]. Climate data were obtained for 2009–2024 to allow lag calculations using the KrigR package in R [22].

**Large-scale (regional) climate indices.** We used the Oceanic Niño Index (ONI), which represents sea surface temperature anomalies in the Niño 3.4 region averaged over three months, as a measure of the El Niño-Southern Oscillation (ENSO). The Dipole Mode Index (DMI), which represents the Indian Ocean Dipole (IOD), was also included. Both indices were obtained using the rsoi package in R [23].

### Wavelet analysis

**Phase angle extraction.** We used Morlet wavelet transformations [24] to extract annual (12-month) phase angles from dengue incidence and climate time series. Prior to transformation, incidence and precipitation data were log-transformed and standardised (z-scores calculated within each province separately to highlight temporal patterns); temperature and relative humidity were standardised without log-transformation. Wavelet analyses were conducted using the biwavelet package in R [25].

**Phase lags between provinces.** We calculated pairwise phase differences (phase lags) between provinces to identify the relative timing of annual dengue cycles. Average phase lags were computed for each province, representing its position in time relative to other provinces. Positive values indicate later peaks (or the start of the annual dengue wave); negative values indicate earlier peaks. We calculated phase lags statistics, including 2.5th-97.5th percentile ranges of pairwise differences.

**Peak month identification.** Peak dengue months were identified as the month with wavelet phase angle closest to zero within each July-June epidemic year. Average peak month across years was calculated using the circular mean to account for the cyclical nature of monthly data.

### Dynamic time warping (DTW) clustering

We used dynamic time warping (DTW) [26] with hierarchical clustering to identify provinces with similar temporal dynamics, independent of the wavelet phase analysis. Missing case counts were imputed using median monthly proportions within each province. Time series were standardised (z-scores) prior to computing pairwise DTW distances.

We performed three clustering analyses at progressively finer geographic scales: 1) all 34 provinces, 2) western Indonesia, including Kalimantan, and 3) western Indonesia excluding Kalimantan (Sumatra and Java-Bali). Optimal cluster numbers were determined using silhouette scores evaluated for $k = 2–15$, with consideration of cluster balance to avoid highly unequal cluster sizes. Clustering was performed using the dtwclust package in R [27].

### Within-region cohesion analysis

To assess whether administrative regions capture temporal dynamics, we calculated within-region cohesion as the mean pairwise DTW distances among provinces within each region. Lower values indicate greater temporal similarity. Provinces whose dynamics deviate from their regional pattern, using z-scores of mean distances to regional neighbours, with a threshold of 1.645 were identified as outliers (corresponding to 90% confidence level).

### Climate-dengue phase relationships

**Phase lag and coherence.** We assessed the temporal relationship between climate and dengue using wavelet-based phase angle analysis. For each province and climate variable, we extracted phase angles at the 12-month period and calculated the phase difference with dengue phase angles at each time point. The mean phase lag (in months) shows

when climate cycles lead or lag dengue cycles, with positive values indicating climate leads dengue. Phase coherence, quantifying the consistency of this timing relationship over the study period, was calculated as:

$$\text{Phase coherence} = 1 - (\text{SD of phase differences} / \pi)$$

where SD is the standard deviation of the wrapped phase differences. Coherence values range from 0 (phase relationship varies randomly over time) to 1 (perfectly consistent timing relationship). Provinces with phase coherence ≥0.85 and positive phase lag were identified as having reliable, predictive climate-dengue timing relationships for early warning applications.

### Distributed lag non-linear models (DLNM)

**Model specification.** We fitted distributed lag non-linear models (DLNM) [28] using the dlnm package in R [29] to quantify lag-exposure-response relationships between local climate variables and dengue incidence for each province. Models used:

- Negative binomial likelihood to handle overdispersion
- Natural cubic spline basis functions for exposure-response and lag-response
- Lag range 0–4 months
- Reference values: median for local climate variables, zero for climate indices
- Adjustment for monthly seasonality

**Cumulative relative risk.** For each climate variable and province, we calculated cumulative relative risk (RR) comparing 90th versus 50th percentile exposure (and 10th versus 50th) at each lag. Two complementary approaches were used to identify optimal lags in DLNM analyses based on statistical significance (95% CI excluding 1) and elevated risk (cumulative relative risk > 1). The 'highest-risk' approach selected the lag with the highest cumulative relative risk among lags with significant elevated risks, while the 'first-significant' approach identified the earliest lag demonstrating significant elevated risk. If there is no lag with statistically significant elevated risk, we chose the lag with the highest cumulative relative risk.

**Phase-DLNM integration.** To link timing (wavelet analysis) with effect magnitude (DLNM), we reported cumulative RR at the wavelet-derived phase lag for provinces with high phase coherence (≥0.85) and positive phase lag. This approach focuses on provinces where climate-based early warning may be feasible, creating continuity between identifying when climate affects dengue and quantifying by how much. We checked if there is a coherence between wavelet-derived lag and DLNM-based lag.

### Statistical software

All analyses were conducted in R version 4.5.2 [30]

## Results

Indonesia reported more than 1.8 million dengue cases during 2010–2024, with marked spatial and temporal heterogeneity across its 34 provinces (Fig 1). The average annual incidence rate varied by about 23-fold across provinces, ranging from 8.3 to 188.7 per 100,000 population. Bali had the highest average annual incidence rate, while eastern provinces (Maluku, Papua) reported consistently lower rates (Fig 1A). National incidence showed distinct multi-annual cycles on top of consistent seasonal patterns, with major outbreak peaks in 2015–2016 and 2023–2024 (z-score of annual incidence

PLOS Neglected Tropical Diseases

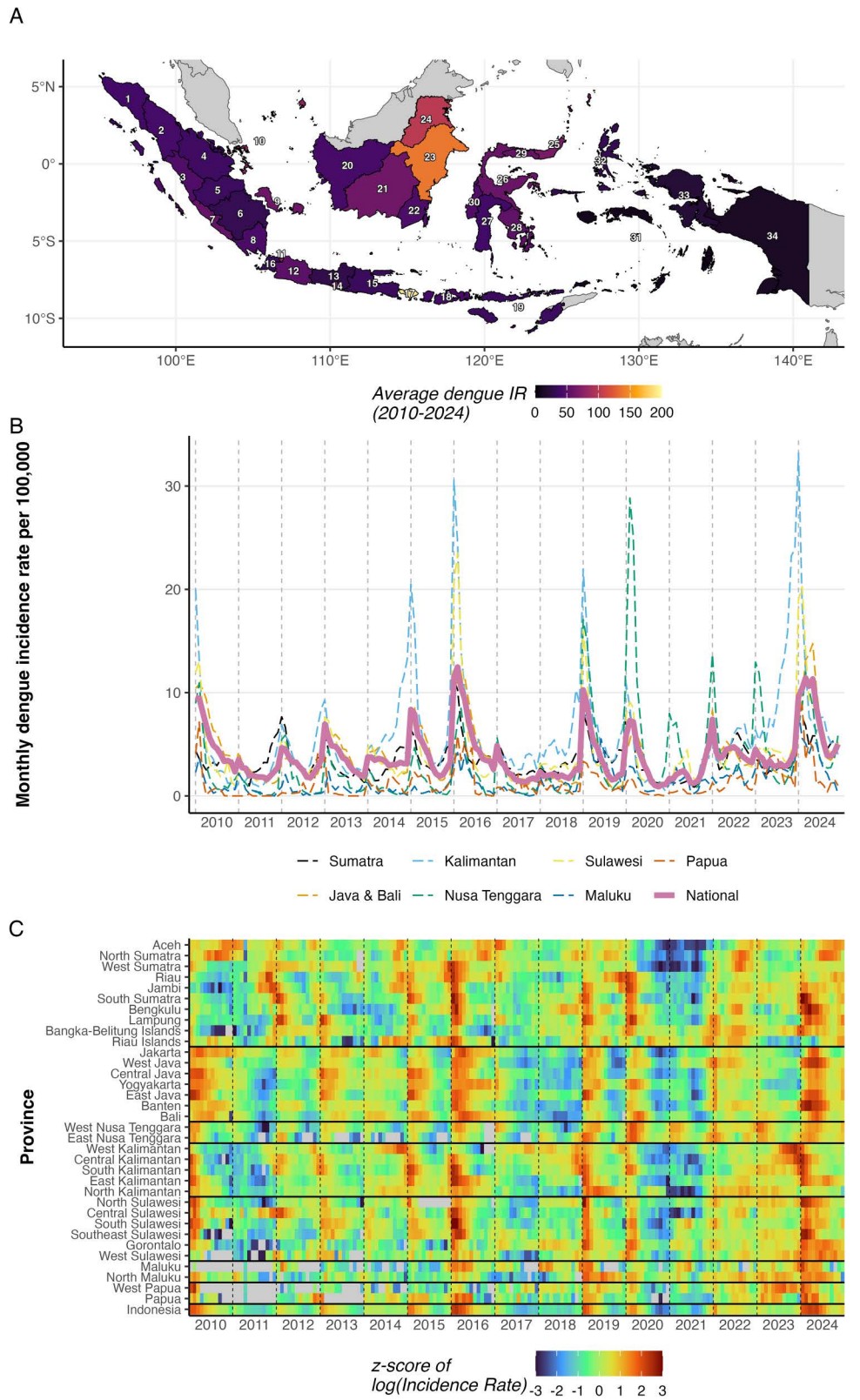

**Fig 1. A) Average annual dengue incidence rate (per 100,000 population) by province over the study period. B)** Monthly incidence rate by geographic region (dashed lines) and nationally (solid line). **C)** Heatmap of z-scored log-transformed monthly incidence rate by province, ordered by

geographic region. Horizontal lines separate regions. Z-scores were calculated within each province. Provinces are ordered as in panel **A**. Names of provinces for each index number are available in **Table A** in S1 Appendix. Administrative boundaries from geoBoundaries [37], CC BY 4.0.

above 1, Fig 1B and Table A in S1 Appendix). The drivers of these multi-annual fluctuations are examined in a later section. Province-level patterns revealed heterogeneous seasonality timing (Fig 1C). Western provinces, especially in Sumatra, showed earlier seasonal peaks, while eastern provinces displayed delayed patterns.

Wavelet analysis revealed a systematic west-to-east gradient in annual dengue wave timing (Fig 2A; mean with 2.5th-97.5th percentile ranges of pairwise differences are available in Fig B in S1 Appendix). Provinces in Sumatra and western Kalimantan showed negative phase lags (averaging peaks 1.4 months earlier than the national average), while Java-Bali and eastern Kalimantan (averaging peaks 0.8 months later) and eastern provinces (averaging peaks 0.1 months later) showed positive phase lags. Median peak outbreak months ranged from October in North Sumatra to April in Jakarta, West Java, Bali, and North Kalimantan (Fig 2B). The gradient's strength varied by geographic scope (Table 1). The west-to-east gradient, as measured by Spearman correlations, was strongest for western provinces only (Sumatra + Java-Bali; median Spearman rho = 0.7, significant across all years), weaker when Kalimantan was included, and weakest across all provinces. Year-to-year variability in peak timing is shown in Figs D and E in S1 Appendix.

Dynamic time warping (DTW) clustering showed that dengue incidence heterogeneity is too complex for simple geographic regionalisation, especially in eastern Indonesia. All 34 provinces required 12 clusters (Fig 3A); restricting to western provinces improved coherence (Fig 3B), with the cleanest structure for Sumatra and Java-Bali only (Fig 3C, silhouette statistics and maximum cluster size for each clustering scenario are available in Tables C, D, and E in S1 Appendix). Temporal similarity within geographical regions varied substantially (Fig 3D and Table 2). The Nusa Tenggara region (two provinces) showed the highest cohesion (mean DTW distance = 82.7), while the Papua region (two provinces) showed the lowest. Only Riau Islands in Sumatra, East Java in Java-Bali, and West Sulawesi in Sulawesi are considered outliers (Table F in S1 Appendix). This within-region heterogeneity was present even among geographically proximate provinces.

We examined climate patterns that may drive the dengue incidence dynamics at the national and provincial levels (Fig 4). Of the two regional climate indices, ONI showed a stronger association with epidemic-year incidence ($\rho = 0.83$; $p < 0.01$) than DMI ($\rho = 0.33$; $p = 0.25$), based on correlations between July-June averaged index values and corresponding annual incidence. Regional variation was observed, with ONI correlations being highest in Kalimantan ($\rho = 0.90$) and Sulawesi ($\rho = 0.83$), whereas DMI exhibited stronger associations than ONI in Maluku ($\rho = 0.60$ vs 0.29) and Nusa Tenggara ($\rho = 0.46$ vs 0.32) (**Table G** in S1 Appendix). The study period encompassed 2 strong El Niño events (2015–2016 and 2023–2024, averaging >1 ONI) and 1 strong La Niña event (2010–2011, averaging < -1 ONI) (Fig 4A). The multi-annual outbreak peaks identified in Fig 1B coincided with strong El Niño conditions. The mean incidence during strong El Niño years was 96% higher than in non-strong El Niño years (77 vs 39 per 100,000). The strongest El Niño (2015–2016, average ONI = 1.81) preceded the 2015–2016 outbreak peak. Local climate showed spatial gradients (Fig 4B-4D). Precipitation exhibited pronounced seasonal wet-dry cycling that was broadly synchronous across western provinces but showed later timing and more heterogeneous patterns in eastern provinces (Fig 4B). Temperature showed weaker seasonality (in western provinces, hotter seasons became slightly cooler due to high precipitation periods), with widespread positive anomalies during El Niño episodes and negative anomalies during La Niña periods (Fig 4C). During strong El Niño years (2015–2016 and 2023–2024), prolonged drought coincided with above-average temperatures, resulting in markedly low humidity across most provinces (Fig 4B-4D). Relative humidity (Fig 4D) largely tracked precipitation patterns with approximately a 1-month delay.

For the following phase and dose-response analyses, we focused on precipitation and temperature, excluding relative humidity due to its high correlation with precipitation and similar but delayed phase patterns. Phase lag between local climate and dengue varied by province (Fig 5A). Positive lags (climate leading dengue) were observed in 19 provinces for

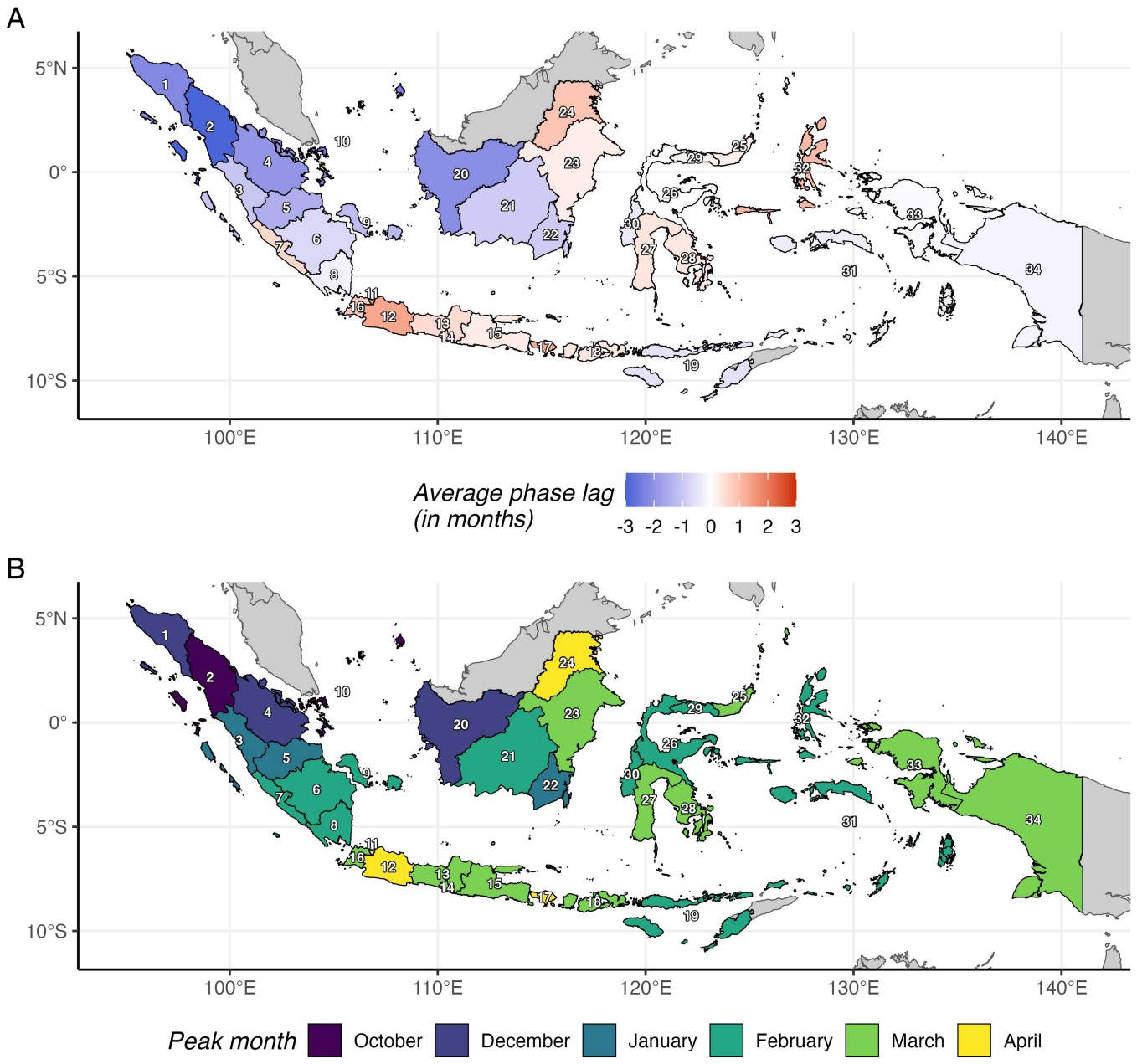

**Fig 2. A) Average phase lag (months) of each province's annual dengue cycle relative to other provinces.** Negative values (blue) indicate earlier peaks; positive values (red) indicate later peaks. **B)** Median peak outbreak month by province, identified using circular statistics over July-June annual periods. Administrative boundaries from geoBoundaries [37], CC BY 4.0.

precipitation and 26 for temperature, ranging from 0.02 to 5.34 months and 0.25 to 5.97 months, respectively. Regional patterns emerge in the phase lags for each climate variable, with some outliers in each region. Phase coherence exceeded 0.85 in 19 provinces for precipitation and 9 for temperature (Fig 5B). This threshold, while necessarily arbitrary, identifies provinces where climate-dengue timing is sufficiently consistent for early warning. Provinces with both positive or

**Table 1. West-to-east gradient in peak timing by province subset.** Spearman rank correlation between province longitude and peak outbreak month (July-June scale) for each epidemic year (2010-2011 to 2023-2024).

| Epidemic year | All provinces (n = 34) | Western + Kalimantan (n = 22) | Western only (n = 17) |
|---|---|---|---|
| 2010-2011 | 0.128 (n.s.) | 0.56 (p < 0.01) | 0.729 (p < 0.01) |
| 2011-2012 | 0.138 (n.s.) | 0.581 (p < 0.01) | 0.802 (p < 0.01) |
| 2012-2013 | 0.134 (n.s.) | 0.6 (p < 0.01) | 0.839 (p < 0.01) |
| 2013-2014 | 0.241 (n.s.) | 0.668 (p < 0.01) | 0.87 (p < 0.01) |
| 2014-2015 | 0.468 (p < 0.01) | 0.576 (p < 0.01) | 0.806 (p < 0.01) |
| 2015-2016 | 0.462 (p < 0.01) | 0.576 (p < 0.01) | 0.83 (p < 0.01) |
| 2016-2017 | 0.417 (p < 0.05) | 0.628 (p < 0.01) | 0.728 (p < 0.01) |
| 2017-2018 | 0.25 (n.s.) | 0.418 (p < 0.10) | 0.648 (p < 0.01) |
| 2018-2019 | 0.389 (p < 0.05) | 0.439 (p < 0.05) | 0.637 (p < 0.01) |
| 2019-2020 | 0.288 (n.s.) | 0.378 (p < 0.10) | 0.592 (p < 0.05) |
| 2020-2021 | 0.326 (p < 0.10) | 0.167 (n.s.) | 0.503 (p < 0.05) |
| 2021-2022 | 0.587 (p < 0.01) | 0.389 (p < 0.10) | 0.696 (p < 0.01) |
| 2022-2023 | 0.419 (p < 0.05) | 0.361 (p < 0.10) | 0.589 (p < 0.05) |
| 2023-2024 | 0.298 (p < 0.10) | 0.288 (n.s.) | 0.521 (p < 0.05) |

Positive correlation indicates westward provinces peak earlier. Significance: *** p < 0.01, ** p < 0.05, * p < 0.10, n.s. = not significant. p-values are nominal and should be interpreted cautiously given potential temporal autocorrelation.

zero lag (after rounding) and high coherence are selected as candidates for climate-based risk prediction (18 for precipitation and 7 for temperature).

For provinces with reliable climate-dengue timing (coherence ≥0.85 and positive or zero lag), high precipitation (90th vs. 50th percentile) was associated with significantly elevated cumulative risk in 11 of 18 qualified provinces at their wavelet-derived optimal lag (Fig 6A). Median cumulative elevated risk for 90th percentile precipitation ranged from 21-95%, and lag timing varied from 0-3 months at statistically significant provinces. For temperature, 3 of 7 qualifying provinces showed significant effects, with one (West Nusa Tenggara) showing reduced cumulative risk at the wavelet-derived optimal lag (Fig 6B). Median cumulative elevated risk for the 90th percentile temperature ranged from 111-236%, and lag timing varied from 0-3 months at statistically significant provinces. The wavelet-derived phase lag and the DLNM-optimal lag (using both 'highest-risk' and 'first-significant' approaches) frequently diverged (**Fig C** in S1 Appendix).

## Discussions

This exploratory study characterised spatiotemporal heterogeneity in dengue incidence across Indonesia's 34 provinces over 15 years (2010–2024) and examined relationships with climate variables. Three principal findings emerged. First, a robust west-to-east gradient in outbreak timing exists in western Indonesia, with provinces in Northern Sumatra peaking earlier than those in other provinces. However, this gradient breaks down in eastern Indonesia. Second, temporal dynamics within administrative regions are heterogeneous, so geographic proximity does not guarantee similar outbreak patterns, suggesting province-specific rather than regional approaches are needed. Third, wavelet-based phase coherence identifies provinces where climate-dengue timing is sufficiently consistent for early warning applications, while DLNM quantifies the magnitude of climate effects in these provinces.

### West-to-east gradient and ecological discontinuity

The west-to-east timing gradient we observed aligns with the progression of the Australian-Asian monsoon system, which moves from northwest to southeast across the Indonesian archipelago [31]. This suggests that provincial differences in

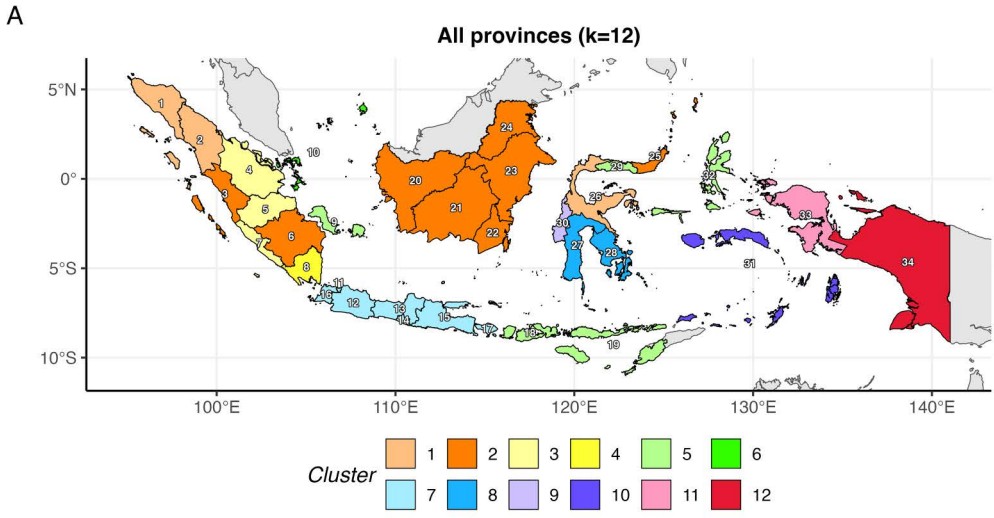

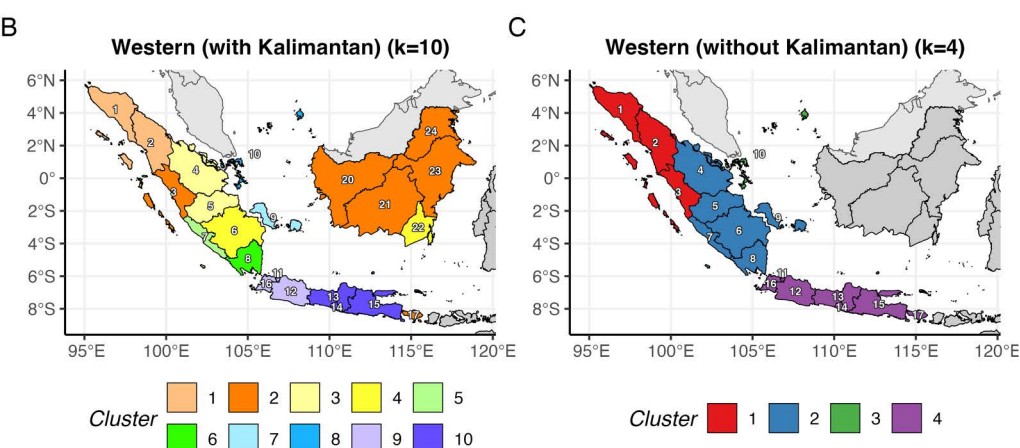

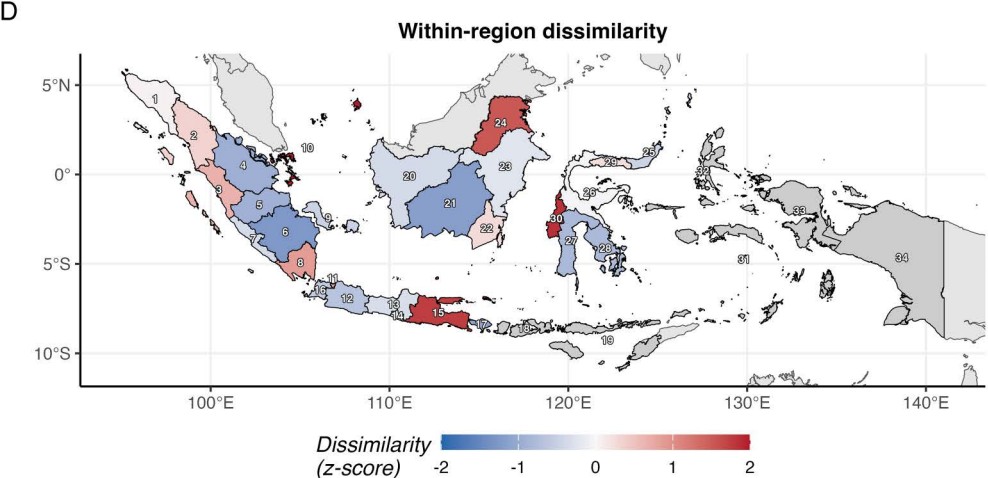

**Fig 3. A) Cluster assignments for all 34 provinces (k = 12). B)** Cluster assignments for western Indonesia including Kalimantan (k = 10); non-included provinces shown in grey. **C)** Cluster assignments for Sumatra and Java-Bali only (k = 4); non-included provinces shown in grey. **D)** Within-region

dissimilarity z-scores based on mean pairwise DTW distance to regional neighbours. Positive values (red) indicate provinces whose temporal dynamics deviate from their region; negative values (blue) indicate high similarity to regional neighbours. Administrative boundaries from geoBoundaries [37], CC BY 4.0.

**Table 2. Within-region temporal pattern cohesion.** Mean pairwise dynamic time warping (DTW) distance between provinces within each geographic region. Lower values indicate more similar temporal dynamics. SD = standard deviation; CV = coefficient of variation. Regions are ranked from most cohesive (lowest mean DTW distance) to least cohesive.

| Region | N provinces | Mean DTW distance¹ | SD | CV |
|---|---|---|---|---|
| Nusa Tenggara | 2 | 82.70 | NA | NA |
| Java & Bali | 7 | 83.34 | 7.68 | 0.09 |
| Kalimantan | 5 | 83.84 | 9.10 | 0.11 |
| Sumatra | 10 | 98.60 | 10.70 | 0.11 |
| Sulawesi | 6 | 105.23 | 10.72 | 0.10 |
| Maluku | 2 | 117.05 | NA | NA |
| Papua | 2 | 123.08 | NA | NA |

¹ Lower mean DTW distance indicates more similar temporal dynamics within the region. CV = coefficient of variation.

peak timing reflect climate-driven seasonality rather than sequential disease spread between provinces. The gradient was strongest when analysis was restricted to Sumatra and Java-Bali (**Table 1**), and weakened substantially when eastern provinces were included. This indicates that the gradient is a robust feature of western Indonesia but breaks down in the eastern archipelago.

Several factors may contribute to the breakdown of the west-to-east gradient in eastern Indonesia. First, climate regimes differ markedly across the archipelago. Aldrian and Susanto identified three distinct climatic regions in Indonesia: a monsoonal region covering southern Indonesia from Sumatra to Timor, a semi-monsoonal/equatorial region in north-western Indonesia, and an anti-monsoonal region encompassing Maluku and northern Sulawesi with opposite seasonal patterns [32]. Eastern Indonesia experiences different climate drivers than western provinces, with ENSO and the Indian Ocean Dipole exerting independent, regionally heterogeneous effects on Indonesian rainfall, and the strength of these associations varies by climate region [33]. Indeed, DMI showed stronger associations with dengue incidence than ONI in Maluku and Nusa Tenggara, whereas ONI dominated in western and central regions. This regional heterogeneity in climate index associations may partially explain the breakdown of the west-to-east gradient in eastern Indonesia. Second, surveillance capacity varies geographically. Eastern provinces consistently report lower dengue incidence, and Indonesia's surveillance system captures only cases presenting to healthcare facilities, potentially underreporting dengue cases, especially in less accessible regions [16]. Notably, the west-to-east gradient was strongest in regions with higher estimated surveillance capacity (Sumatra and Java-Bali), suggesting this pattern may reflect genuine epidemiological dynamics rather than reporting artifacts. Finally, while dengue vectors occur throughout the archipelago [34], serotype distribution and immunity patterns in the population differ between regions and cannot be generalised nationally [35]. Disentangling these climatic, surveillance, and immunological factors requires targeted studies in eastern Indonesia and is beyond the scope of this exploratory analysis.

## Heterogeneity within geographical regions

Our clustering analysis revealed that geographical regions are poor proxies for dengue temporal dynamics. Despite geographic proximity, provinces within the same region showed substantial heterogeneity in outbreak timing and amplitude (Fig 3D and **Table 2**). This has practical implications: regional-level policies may fail to account for province-specific

**Fig 4. A) Time series of the Oceanic Nino Index (ONI) and Dipole Mode Index (DMI).** Horizontal reference lines indicate thresholds for moderate (+-0.5) and strong (+-1.0, +-1.5, +-2.0) events. **B-D)** Heatmaps of z-scored monthly precipitation **B)**, temperature **C)**, and relative humidity **D)** by

province, ordered by geographic region. Horizontal lines separate regions. Z-scores were calculated within each province. Dashed vertical lines indicate January of each year.

dynamics. The identification of temporal outliers, provinces whose dynamics deviate from their regional neighbours, high-lights where province-specific investigation may be warranted, as these outliers may reflect unique local factors such as serotype circulation history [35,5] or vector control programme effectiveness.

### Climate-dengue relationships at two timescales

Climate influences dengue at two distinct timescales, each with different implications for preparedness.

At the multi-annual scale, El Niño events were associated with elevated outbreak risk. ONI showed a significant positive correlation with epidemic-year incidence (Spearman $\rho = 0.83$, p < 0.01), while DMI did not ($\rho = 0.33$, p = 0.25). Mean incidence during strong El Niño years was 96% higher than during non-strong El Niño years (77 vs 39 per 100,000). This association, while not establishing causality, suggests that ENSO monitoring can inform strategic preparedness such as budget allocation, supply chain planning, and workforce training months in advance of anticipated high-risk periods.

At the annual scale, local climate variables showed province-specific timing relationships with dengue. We focused on local climate rather than large-scale indices for the phase and dose-response analyses because they operate at different timescales: ONI/DMI capture multi-annual ENSO/IOD cycles, whereas local climate exhibits annual seasonality that aligns with the 12-month dengue cycles extracted by wavelet analysis. Local climate also provides actionable signals for tactical interventions at 1–4 month lead times, while ENSO serves strategic planning at longer horizons.

Phase coherence analysis identified provinces where climate-dengue timing was sufficiently consistent for potential early warning applications. Precipitation showed broader utility than temperature, with 18 qualifying provinces (coherence >=0.85, positive or zero lag) compared to 7 for temperature, and 11 significant dose-response associations versus 3. This asymmetry may reflect precipitation's more direct mechanistic pathway to dengue transmission through breeding site creation, while temperature effects on vectorial capacity may be modulated by other factors that were not explained by the simple model we employed.

Notably, 16 provinces had low phase coherence (<0.70) for all climate variables examined, suggesting that factors other than direct climate effects, such as serotype dynamics, population immunity, or reporting inconsistencies, may dominate their outbreak patterns. For these provinces, climate-based early warning may be less effective than alternative approaches that account for other potential factors.

### Complementarity of wavelet and DLNM approaches

The wavelet-derived phase lag and the DLNM-optimal lag frequently diverged (**Fig C** in S1 Appendix) as expected, because they measure fundamentally different aspects of the climate-dengue relationship. Wavelet phase lag captures when annual cycles align under a sinusoidal assumption, while DLNM identifies the lag with strong or the strongest dose-response effect, allowing non-linear relationships. This divergence supports using both approaches as complementary tools: phase analysis screens for reliable timing relationships, while DLNM quantifies the magnitude of the effects. For West Nusa Tenggara, DLNM revealed a protective effect of high temperature (decreased cumulative RR), illustrating how dose-response analysis may uncover relationships not apparent from timing analysis alone.

For operational early warning applications, these complementary methods provide different actionable information. Phase coherence (wavelet) screens which provinces have reliable climate-dengue timing relationships worth

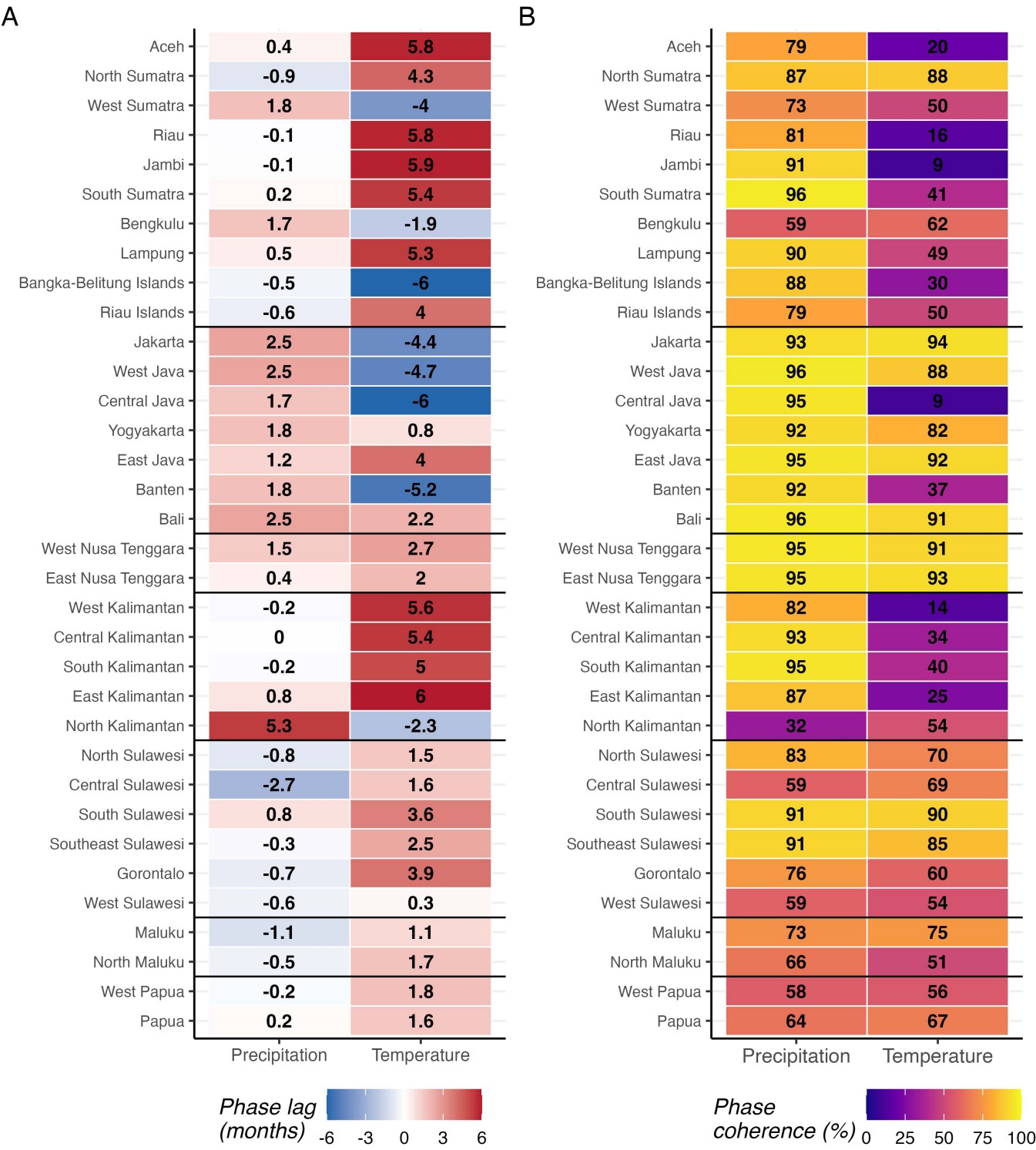

**Fig 5. A) Wavelet-derived phase lag (months) between local climate variables and dengue incidence.** Positive values (red) indicate climate leading dengue; negative values (blue) indicate dengue leading climate. **B)** Phase coherence (%) quantifying the consistency of the climate-dengue timing relationship over the study period. Higher values indicate more reliable timing relationships. Provinces are ordered by geographic region with horizontal lines separating regions. Values are displayed within cells.

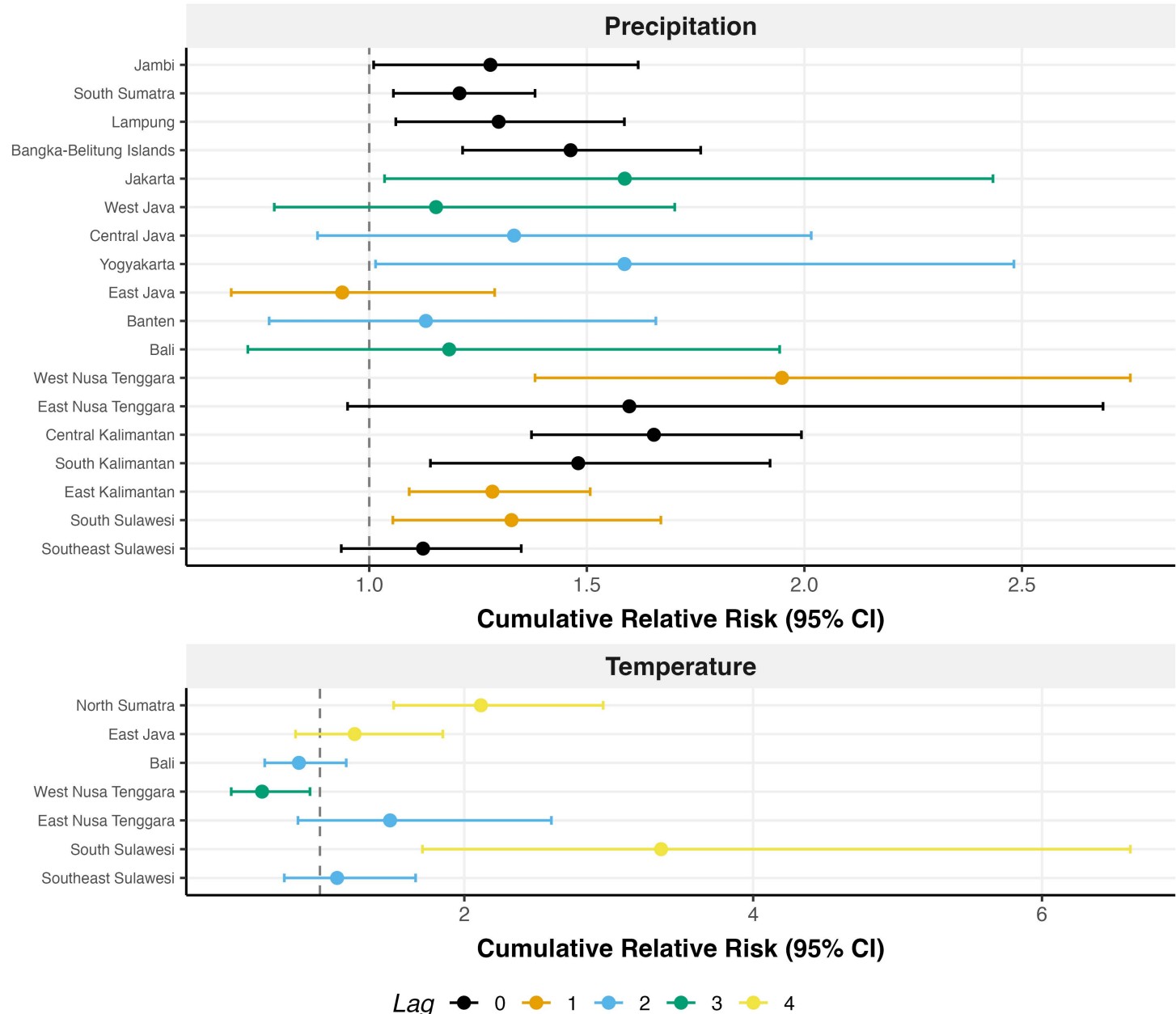

**Fig 6. Forest plots showing cumulative relative risk (RR) with 95% confidence intervals for 90th versus 50th percentile exposure to precipitation A) and temperature B).** Only provinces with phase coherence >= 0.85 and positive or zero phase lag are included. Point colour indicates the wavelet-derived lag (months) at which cumulative RR was evaluated. Vertical dashed line indicates RR = 1 (no effect).

monitoring, while DLNM optimal lag identifies when the strongest lag-dose-response signal occurs for intervention timing. When wavelet and DLNM lags diverge, DLNM lag may be more directly actionable for tactical decision-making as it captures non-linear exposure-response relationships, though prospective operational research is needed to validate this approach.

## Implications for early warning

Our findings support a two-tier approach to dengue early warning. The first tier, based on ENSO monitoring, provides strategic warning at 2–6 + month horizons for outbreak years, applicable nationally. The second tier, based on local climate monitoring, provides tactical warning at 1–4 month horizons for seasonal peaks in provinces with high phase coherence. The phase coherence metric may serve as a screening tool to identify where climate-based prediction is most feasible.

Indonesia's dengue control is largely decentralised, with provincial and district health offices implementing interventions based on local trends, while the national government provides coordination through initiatives such as the 2021–2025 National Strategic Plan for Dengue Control [36]. Our findings support this structure but suggest that inter-provincial coordination, enhanced by coordination and monitoring at the national level, could optimise resource deployment by enabling earlier-peaking provinces to inform preparedness in later-peaking neighbours.

To illustrate, consider North Sumatra (early-peaking, October) and Central Java (later-peaking, February), both with high precipitation phase coherence. In North Sumatra, heightened September rainfall would signal October risk; in Central Java, November–December signals would inform January–February preparedness. This staggered approach could enable more efficient resource allocation than uniform national timing, though prospective validation remains essential.

## Limitations

Several limitations should be acknowledged. First, this study is exploratory, aimed at characterising heterogeneous patterns rather than testing causal hypotheses. The associations we report do not establish causality and require prospective validation before operational use.

Second, dengue surveillance quality varies across Indonesia, with eastern provinces likely experiencing greater under-reporting than Java and Sumatra. The consistently lower incidence and phase coherence in eastern provinces may partly reflect data limitations rather than true epidemiological differences. Indonesia's dengue surveillance has used consistent clinical case definitions for dengue hemorrhagic fever throughout the study period [15,16], though laboratory confirmation capacity and reporting completeness have likely improved over time, particularly in western provinces.

Third, the COVID-19 pandemic (2020–2021) likely affected both dengue transmission and surveillance reporting. However, our 15-year study period provides a decade of pre-pandemic data, reducing the influence of pandemic-related anomalies on our findings. Sensitivity analyses excluding 2020–2021 showed that phase lag estimates, phase coherence values, and DLNM associations were qualitatively similar to the full period results (**Tables H, I,** and **J** in S1 Appendix).

Fourth, our DLNM analyses modelled each climate variable separately, adjusting only for seasonality, which does not account for correlations between climate variables or potential confounding. Similarly, while we demonstrated strong associations between ENSO (ONI) and multi-annual outbreak patterns, we did not model these large-scale climate indices within the DLNM framework. Such indices may improve predictive power in multivariate models that adjust for the effects of local climate variables such as monthly rainfall, though disentangling these complex delayed effects warrants dedicated investigation. Additionally, the Spearman correlation between ONI and epidemic-year incidence is based on a small sample size (n = 14 epidemic years), and temporal autocorrelation in monthly time series may reduce effective sample sizes and inflate apparent significance of phase-lag correlations. The p-values reported in Table 1 should be interpreted as nominal values given potential autocorrelation structure in the data.

Fifth, the phase coherence threshold of 0.85, while chosen to identify provinces with reliable timing relationships, is necessarily arbitrary. Sensitivity analysis across thresholds from 0.70 to 0.90 showed that the number of provinces with significant DLNM effects remained relatively stable, indicating that our conclusions are not highly sensitive to this choice (**Table K** in S1 Appendix).

Sixth, we lacked serotype-specific data, a major driver of outbreak dynamics through immunity patterns. Studies have shown that DENV serotype distribution in Indonesia varies substantially between cities and cannot be generalised

nationally [35]. This province-level variation in serotype circulation could explain some heterogeneity in climate-dengue relationships. Finally, province-level analysis may mask important sub-provincial variation, particularly in large provinces where district-level dynamics could differ substantially.

## Conclusions and future directions

Indonesia's dengue-climate relationships are characterised by substantial heterogeneity that precludes uniform national prediction approaches. However, this heterogeneity is structured: a robust timing gradient exists in western Indonesia, and phase coherence analysis helps identify provinces where climate-based prediction is feasible.

Future research should prioritise: 1) prospective validation of phase coherence as a predictor of early warning system performance; 2) strengthening surveillance in eastern Indonesia to distinguish true heterogeneity from data limitations; 3) incorporating serotype surveillance and seroprevalence surveys to understand immunity-driven dynamics; and 4) operational testing of province-specific early warning systems in high-coherence provinces.

## Supporting information

**S1 Appendix. Table A:** Province names and their corresponding numerical indices fro reference to all maps in the main figures. **Table B:** Annual national dengue incidence by epidemic year (July-June), with z-scores relative to the study period mean. Epidemic year "2015-16" represents July 2015 to June 2016. Major outbreak years are defined as z-score > 1 (more than one standard deviation above the mean). **Table C**: Silhouette statistics for different numbers of clusters (k = 2 to k = 15) for all 34 provinces. The selected number of clusters (k = 12) accounts for both silhouette statistics and balance in cluster sizes. **Table D**: Silhouette statistics for different numbers of clusters (k = 2 to k = 15) for western provinces including Kalimantan (22 provinces). The selected number of clusters is k = 10. **Table E**: Silhouette statistics for different numbers of clusters (k = 2 to k = 15) for western provinces only (Sumatra and Java-Bali, 17 provinces). The selected number of clusters is k = 4, representing the cleanest cluster structure. **Table F**: Mean DTW distance from each province to other provinces in the same geographic region, with z-scores calculated within each region. Provinces with |z| > 1.65 are flagged as temporal outliers whose dynamics deviate substantially from their regional pattern. **Table G**: Spearman correlations between large-scale climate indices (ONI and DMI) and dengue incidence by geographic region, using epidemic year aggregation (July-June). **Table H**: Sensitivity analysis (COVID-19): comparison of province-level dengue phase lags between the full study period (2010–2024) and the pre-pandemic period (2010–2019). **Table I**: Sensitivity analysis (COVID-19): comparison of climate-dengue phase coherence and qualifying status (coherence ≥ 0.85, phase lag ≥ 0) between the full study period (2010–2024) and the pre-pandemic period (2010–2019), for precipitation and temperature. **Table J**: Sensitivity analysis (COVID-19): comparison of DLNM-derived cumulative relative risks at wavelet-derived lags between the full study period (2010–2024) and the pre-pandemic period (2010–2019). Only provinces qualifying in the full period analysis are shown. **Table K**: Sensitivity analysis (phase coherence threshold): number of qualifying provinces and number showing statistically significant DLNM effects across coherence thresholds ranging from 0.70 to 0.90. The threshold used in the main analysis (0.85) is highlighted. **Fig A**: Geographic regions of Indonesia used for regional stratification throughout the study. **Fig B**: Province-level dengue phase lags relative to all other provinces, with 2.5th–97.5th percentile ranges of pairwise phase differences. Positive values indicate provinces with later epidemic peaks; negative values indicate earlier peaks. Error bars represent the spread of phase differences across all province pairs, not statistical confidence intervals. **Fig C**: Comparison between wavelet-derived phase lags and DLNM-based optimal lags for precipitation and temperature. A) DLNM lags selected using the highest-risk approach (lag with maximum cumulative RR). B) DLNM lags selected using the first-significant approach (earliest lag with significant elevated risk). The red dashed line indicates perfect agreement. Divergence between methods is expected as wavelet phase assumes sinusoidal annual cycles while DLNM allows non-linear dose-response relationships. **Fig D**: Year-to-year variability in peak outbreak timing across all 34 provinces, ordered by longitude (west to east). Each cell shows the peak month for that province in that

epidemic year (July-June). Grey cells indicate anomalous periods excluded due to weak annual periodicity. **Fig E**: Year-to-year variability in peak outbreak timing for western provinces only (Sumatra and Java-Bali), ordered by longitude (west to east). Despite stronger regional coherence than eastern provinces, substantial inter-annual variation in peak timing persists.
(PDF)

## Acknowledgments

We thank the Arbovirus Working Team at the Ministry of Health of the Republic of Indonesia for providing access to the surveillance data and technical support.

## Author contributions

**Conceptualization:** Bimandra A Djaafara, Swapnil Mishra.

**Data curation:** Fadjar SM Silalahi, Agus Handito, Burhannudin Thohir, Desfalina Aryani.

**Formal analysis:** Bimandra A Djaafara.

**Funding acquisition:** Swapnil Mishra.

**Methodology:** Bimandra A Djaafara, Swapnil Mishra.

**Writing – original draft:** Bimandra A Djaafara.

**Writing – review & editing:** Iqbal RF Elyazar, Fadjar SM Silalahi, Asik Surya, Agus Handito, Burhannudin Thohir, Desfalina Aryani, Mushtofa Kamal, Aditya L Ramadona, Dyana Gunawan, Hipokrates, Anzala Khoirun Nisa, Edi Prianto, Iriani Samad, Agus Sugiarto, Kimberly Fornace, Hannah E Clapham, Nuno R Faria, Swapnil Mishra.

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
