## [Decision Letter · Decision Letter 0]

3 Dec 2025

Dengue transmission heterogeneity across Indonesia’s archipelago: climate-driven spatiotemporal patterns and policy implications

Dear Dr. Djaafara,

Thank you for submitting your manuscript to PLOS Neglected Tropical Diseases. After careful consideration, we feel that it has merit but does not fully meet PLOS Neglected Tropical Diseases's publication criteria as it currently stands. Therefore, we invite you to submit a revised version of the manuscript that addresses the points raised during the review process.

Please submit your revised manuscript within by Feb 01 2026 11:59PM. If you will need more time than this to complete your revisions, please reply to this message or contact the journal office at plosntds@plos.org. Please include the following items when submitting your revised manuscript:

We look forward to receiving your revised manuscript.

Kind regards,

David Safronetz, Ph.D.

Section Editor

David Safronetz

Section Editor

Shaden Kamhawi

co-Editor-in-Chief

Paul Brindley

co-Editor-in-Chief

**Journal Requirements:**

At this stage, the following Authors/Authors require contributions: Bimandra A Djaafara, Iqbal RF Elyazar, Fadjar SM Silalahi, Asik Surya, Agus Handito, Burhannudin Thohir, Desfalina Aryani, Mushtofa Kamal, Aditya L Ramadona, Dyana Gunawan, Hipokrates Hipokrates, Anzala Khoirun Nisa, Edi Prianto, Iriani Samad, Agus Sugiarto, Kimberly Fornace, Hannah E Clapham, Nuno R Faria, and Swapnil Mishra. Please ensure that the full contributions of each author are acknowledged in the "Add/Edit/Remove Authors" section of our submission form.

Potential Copyright Issues:

i) Figures 1A, 1B, 2B, 3, and 5A-5C. Please (a) provide a direct link to the base layer of the map (i.e., the country or region border shape) and ensure this is also included in the figure legend; and (b) provide a link to the terms of use / license information for the base layer image or shapefile. We cannot publish proprietary or copyrighted maps (e.g. Google Maps, Mapquest) and the terms of use for your map base layer must be compatible with our CC BY 4.0 license.

**Reviewers' Comments:**

Reviewer's Responses to Questions

**Key Review Criteria Required for Acceptance?**

**Methods**

-Are the objectives of the study clearly articulated with a clear testable hypothesis stated?

-Is the study design appropriate to address the stated objectives?

-Is the population clearly described and appropriate for the hypothesis being tested?

-Is the sample size sufficient to ensure adequate power to address the hypothesis being tested?

-Were correct statistical analysis used to support conclusions?

-Are there concerns about ethical or regulatory requirements being met?

Reviewer #1: (No Response)

Reviewer #2: (No Response)

Reviewer #3: I have concerns about the validity of adjusting dengue incidence for model-based estimates of 'surveillance capacity', the derivation of the 'urban population density' metric, and the use of linear correlation to report associations that are clearly non-linear; see Major Comments below

**Results**

-Does the analysis presented match the analysis plan?

-Are the results clearly and completely presented?

-Are the figures (Tables, Images) of sufficient quality for clarity?

Reviewer #1: (No Response)

Reviewer #2: (No Response)

Reviewer #3: I have outlined my concerns with the presentation and interpretation of results in Major Comments, below.

**Conclusions**

-Are the conclusions supported by the data presented?

-Are the limitations of analysis clearly described?

-Do the authors discuss how these data can be helpful to advance our understanding of the topic under study?

-Is public health relevance addressed?

Reviewer #1: (No Response)

Reviewer #2: (No Response)

Reviewer #3: I have outlined my concerns with the discussion and conclusions in Major Comments, below.

**Editorial and Data Presentation Modifications?**

Reviewer #1: (No Response)

Reviewer #2: (No Response)

Reviewer #3: (No Response)

**Summary and General Comments**

Reviewer #1: Dear Editor,

It is with gratitude that I acknowledge the opportunity to undertake a review of this manuscript, which explores the spatiotemporal heterogeneity and climate synchronization of dengue incidence in Indonesia during the period 2016–2024. The study has compiled a valuable provincial panel and presented clear descriptive patterns, including west–east phase progression and synchronization with El Niño events. However, from the perspective of infectious disease epidemiology and surveillance, there are several substantive issues that limit causal interpretability, policy relevance, and the robustness of the reported associations. The subsequent section will present a summary of my major questions posed to the authors.

1. The manuscript does not provide explicit exposure–response functions linking precipitation, temperature, and relative humidity to dengue incidence, nor does it quantify lagged, non-linear effects with effect sizes and uncertainty. In the absence of dose–response curves and credible intervals, it is challenging to evaluate the magnitude, configuration, or limits of climate-related impacts. Moreover, it is challenging to identify critical lag windows for transmission-relevant processes (e.g. mosquito development, viral extrinsic incubation, human–vector contact). Furthermore, the translation of findings into early warning triggers and actionable risk communication is impeded.

2. The correlation, phase, and coherence analyses do not specify how confounding structures were handled, particularly long-term trends, seasonality, province-specific baseline risk, surveillance intensity and reporting changes (including disruptions caused by the emergence of the Coronavirus), urbanization dynamics, and broader climate modes such as El Niño Southern Oscillation (ENSO) that co-vary with both local climate and incidence. The question of whether the time series were prewhitened or residualized prior to correlation in order to mitigate autocorrelation and non-stationarity remains unresolved. Furthermore, the extent to which any reported associations persist after adjustment for these confounders in a unified inferential framework is unclear.

3. It is evident that the manuscript is not able to provide a convincing demonstration that the observed spatial and temporal patterns are not artefacts of detection variance. This is primarily due to the documented heterogeneity in surveillance capacity across provinces and the pandemic-era perturbation to reporting. It is imperative for the reader to comprehend the reliance of primary inferences on surveillance-adjusted outcomes or models that explicitly adjust for surveillance proxies. Furthermore, it is essential to elucidate the treatment of pandemic years and the robustness of key results when excluding or down-weighting periods of clear reporting disruption.

4. It is possible that the study period encompasses alterations in case definitions, laboratory confirmation rates, and reporting workflows, which have the potential to impact the comparability of the data both longitudinally and across provinces. However, the manuscript does not provide documentation of these shifts nor does it elucidate the manner in which they were addressed in the analytical process. Absent a surveillance policy timeline and corresponding model adjustments, trends and cross-provincial differences may blend epidemiologic signal with administrative variation.

5. The provenance, resolution, preprocessing, and provincial aggregation of climate data are not sufficiently detailed to judge measurement validity or reproducibility, and the manuscript does not assess sensitivity to alternative climate products or aggregation schemes. In light of the established discrepancies between reanalysis and satellite-derived datasets, particularly with regard to precipitation and humidity, it is imperative that the results are demonstrated to be consistent across a range of data selections and grid-to-province weighting methodologies.

6. The central result of the manuscript is the demonstration of synchronisation with ENSO; however, the document does not clearly disentangle basin-scale ENSO effects from local precipitation/temperature/humidity associations. This separation is crucial to avoid attribution bias. In the absence of a framework that considers ENSO in conjunction with local exposures, and that evaluates interaction or mediation structures, it is challenging to ascertain whether the observed timing reflects large-scale forcing, local amplification, or a combination of both.

7. Statistical inference in the presence of strong autocorrelation and potential non-stationarity is not adequately justified. The manuscript should elucidate the manner in which temporal dependence was addressed to circumvent the exaggeration of significance, the diagnostics (e.g., ACF/PACF, stationarity tests) that substantiate the adequacy of the model, and the validation of conclusions through approaches that explicitly model serial correlation or utilise residualised series for correlation-based summaries.

8. The phase and coherence analyses lack detail on key methodological choices (e.g. window lengths, spectral or wavelet parameters, frequency bands) and provide limited treatment of uncertainty. In the absence of quantification, both precision and sensitivity to analytic settings, it is challenging to ascertain which phase relationships are stable and which may be artefacts of parameterisation or data noise.

9. The definition of incidence denominators and the handling of population dynamics are not fully transparent, particularly with respect to population growth, migration, and urbanisation over 2016–2024. It is imperative to employ clear denominator specification and population adjustments to interpret changes in incidence as epidemiological rather than purely demographic, and to assess comparability across provinces with divergent demographic trajectories.

10. The manuscript reports relationships involving urban density, but there is a lack of sufficient articulation of the mechanistic links to dengue transmission, and of evaluation of heterogeneity in climate–dengue associations across urbanisation gradients. However, without examining the effect modification of urban form, container habitats, and human contact patterns, the role of urbanisation remains descriptive and its implications for targeted control strategies underdeveloped.

11. It is important to note that the manuscript does not undertake a formal attribution analysis to determine the relative contributions of human mobility versus climate variability to dengue transmission dynamics. This is despite both being principal drivers of spatiotemporal spread. In the absence of mobility data integration (e.g., inter-provincial flows from mobile devices, transport networks, or holiday schedules) and a framework to apportion timing and amplitude of outbreaks, it is unclear whether observed phase progression and synchronization are predominantly climate-forced, mobility-mediated, or an interaction of the two. This gap limits the manuscript's capacity to inform interventions that are sensitive to movement patterns (e.g., travel advisories, vector control staging, or cross-jurisdictional coordination).

Reviewer #2: This manuscript draws on Indonesia’s provincial dengue surveillance data (2016–2024), combines ERA5-Land climate variables, urbanization metrics, and ENSO (ONI), and uses wavelet phase analysis to depict pronounced spatiotemporal heterogeneity, a west-to-east gradient in the timing of annual peaks, and associations with the multi-year periodicity of El Niño. It also attempts to discuss how differences in surveillance capacity and urban density may influence the conclusions.

1. The current data availability statement—“available upon request from Indonesia’s Ministry of Health according to policy”—may not meet PLOS’s requirement that underlying data be fully available without undue restriction. At a minimum, province-by-month aggregated incidence time series should be openly shared.

2. The manuscript adopts the surveillance “capacity” index (0–1) from Lim et al. and uses it to “adjust” incidence, but the adjustment procedure is not specified (divide by the index? linear rescaling?). Please make the adjustment formula and assumptions explicit (linear/log scale? restricted to urban population?), propagate uncertainty in the capacity estimates via Bayesian methods or bootstrap, and report confidence/posterior intervals for the adjusted incidence. In a multivariable framework, include surveillance capacity as a covariate rather than solely as a scaling factor, and assess robustness.

3. Using ONI alone risks missing the role of the Indian Ocean Dipole (IOD), which is tightly coupled to Indonesian rainfall. Please add DMI/MEIv2 in the main text or supplement, and compare their independent and interactive effects with ONI. To characterize nonlinear and lagged climate–incidence relationships, use DLNM (distributed lag nonlinear models) or GAM-DLNM to quantify exposure–response curves and cross-validate against the wavelet-phase results; at least report effect sizes (phase alignment ≠ effect magnitude). The description in Figure 4 of a 2–3-month lag between ONI and incidence should be quantified with point estimates and intervals.

4. “Urban density” is defined as the ratio of urban population share to urban land share, which is not dimensionally intuitive and is sensitive to small denominators; “urbanization level” and “density” may also be collinear with surveillance capacity/economic development. Use a standard population density measure (persons per km²) from GHS-POP or WorldPop; in multivariable (spatial) regression, include covariates such as surveillance capacity, urbanization, population density, elevation, road density, WASH proxies, and healthcare access. Subject simple bivariate findings (Fig. 5D/E) to robustness checks (replace the density metric; stratify by Java vs. non-Java).

5. Population for 2016–2024 is linearly interpolated from the 2010 and 2020 censuses, ignoring shifts in age structure and migration; COVID-19 also altered care-seeking and surveillance. These issues should be addressed or sensitivity-tested.

6. The phase gradient in Figure 2 is interpreted as differences in climate-driven optimal transmission timing rather than inter-provincial spread. Test the gradient formally via circular–linear regression of phase on longitude/latitude (or monsoon onset dates), and use dynamic time warping (DTW) and/or cross-sectional lag regression to distinguish a common exogenous climate driver from time-ordered propagation between neighboring provinces.

7. Given the anomalously low incidence in 2020, explicitly include intervention/indicator variables in the main models (e.g., Google Mobility/policy stringency).

8. Add a reproducible worked example: choose two contrasting regions (e.g., Sumatra’s west coast vs. Central Java) and present a flowchart and timeline that links climate - phase - an intervention window (lead time 1–2 months) to a concrete package of measures (risk communication, source reduction, ULV timing, early clinical detection). Quantify a plausible range of case reductions (a simple historical counterfactual simulation is acceptable).

Reviewer #3: The authors analyse spatial and temporal patterns in province-level monthly dengue incidence in Indonesia over a 9 year period (2016-2024), a associations with climatic variables and urbanicity. Variability is reported across provinces in the magnitude and temporal variability in dengue incidence, as well as in the timing of epidemic cycles and climate associations. I have several major concerns with the methodology and the interpretation of the analyses, as follows:

MAJOR COMMENTS

1. My major concern is that the analyses reported in this manuscript are all based on notified province-level dengue incidence that has been adjusted for variability in ‘surveillance capacity’ inferred from the output of a previously published global model, but without sufficient demonstration of the validity of those estimates at a subnational level in Indonesia. Figure 5C shows the province-level values for inferred surveillance capacity indicating a high level of variability throughout Indonesia. It is not clear what the effect of adjusting for this (highly variable) inferred surveillance capacity was on province-level dengue incidence rates and the resulting analyses - except what can be inferred from Figure 5D, which shows a dramatic change in the range of province-level adjusted versus unadjusted incidence rates. Presuming that all of the results reported prior to Fig 5 are based on adjusted dengue incidence, the authors should include either in the main manuscript or in supplementary material i) additional validation of the subnational estimates of ‘surveillance capacity’ for Indonesia, ii) a comparison between the unadjusted province-level dengue incidence and the adjusted incidence used in the analyses, either in a table or as a supplementary figure equivalent to Figure 1A (with a comment on this comparison included in the main manuscript text), and iii) a sensitivity analysis, testing whether the associations between climate variables and dengue incidence are different when the unadjusted incidence rates are used.

2. The calculation of ‘urban population density’ described at lines 128-30 of the Methods, and reported in Figure 5B and at lines 267-71, is very unclear in terms of what this metric actually represents and why is it largely the inverse of the simple urbanicity metric. The association between ‘urban population density’ and dengue incidence is oversimplified and even misrepresented in the discussion at lines 300-2 “After adjusting for urban surveillance capacity, we found a significant correlation between provinces with high urban population density and elevated dengue incidence rates” and 370 “we found a positive relationship between adjusted incidence and urban population density”, because: i) this secondary urbanicity metric does not represent ‘urban population density’ as typically understood (i.e. average population per km2 in urban areas); ii) the adjustment for ‘urban surveillance capacity’ drastically changed both the province-level incidence values, without adequate validation of and rationale for this adjustment method, as per above, and iii) the association between adjusted incidence and ‘density of urban’ in Figure 5E is clearly not a linear correlation, the values of ‘density of urban’ are highly skewed and reporting a linear correlation coefficient is not appropriate.

3. I am not convinced that the phase lag analysis as currently reported is informative about spatial-temporal patterns in dengue epidemic cycles across Indonesia. The methods state that pairwise phase lags were calculated - it is unclear whether this produced one overall phase lag for each province pair across the 9 years, or separate phase lags for each of the 9 annual cycles - then averaged for each province. Figure 2 shows the median phase lag for each province, but it is clear from the error bars in Fig 2A that there is a huge amount of variability in the relative timing of dengue cycles among provinces and/or year-to-year. The min and max values of the error bars are very similar across the majority of provinces, so it is hard to see how the median value is particularly informative. It would be much more informative to report explicitly the year-to-year variability in the relative timing of province-level dengue cycles, to understand whether the west-to-east progression in the timing of the dengue season is consistently observed, or whether there is variability in this pattern year to year.

4. The authors interpret the coefficient of variation in province-level dengue incidence or climate variables as indicating the degree of seasonality, however the CoV also reflects interannual variability across the 9 years as well as within-year seasonality. This should be considered in the interpretation of the bivariate maps.

5. The authors conclude that dengue control programs should consider province-specific timing of interventions rather than uniform national timing, to reflect these spatial differences in dengue outbreak cycles. How does this correspond to the current strategy in Indonesia - are public health and vector control campaigns actually centralised and implemented with uniform timing, or are they already decentralised to subnational health authorities to implement in response to local disease trends?

6. The discussion should be considerably shortened and tightened up to reduce overinterpretation of the analyses and speculative, sometimes contradictory statements. For example line 308 “This pattern reflects climate-driven differences in optimal transmission timing rather than disease spread between provinces.” contradicts line 402: “A granular understanding of human mobility and connectivity across the country could further contribute to characterising how movement between regions may drive the spatial spread of dengue.”

7. For all maps, move the labels off to the side of provinces in Java, as they are blocking visibility of the colour shading.

8. Figure 1C: the figure legend states that the heatmap displays ‘log-transformed and scaled monthly incidence rate’, which appears to be a different scale than the absolute (annual) dengue incidence rate shown in Figure 1A. This is currently misleading, as the same colour range is used in Fig 1A and 1C yet the scale is different. Please include a separate colour range legend for Fig 1C. Also clarify in the figure legend and in the Methods section how the monthly incidence was scaled: presumably this means normalising to zero mean and unit variance - was this normalisation done across all months and provinces, not across all months within each province? This needs to be stated explicitly.

MINOR COMMENTS

9. There is a typo in the Figure 1A legend, ‘2016-2024’ (not 2014).

10. Line 66-9: Update the numbers and reference to include all of 2024, not just the first quarter.

11. Line 97: The citation of reference 13 as the source for the dengue case notification data used in this analysis does not appear to be correct.

12. Line 160: ‘Each variable was categorised into three groups using each variable’s quartile…’ Shouldn’t this be four groups (i.e. quartiles)?

13. Supplementary Files 2 and 3 need figure legends.

PLOS authors have the option to publish the peer review history of their article (what does this mean? ). If published, this will include your full peer review and any attached files.

**Do you want your identity to be public for this peer review?** For information about this choice, including consent withdrawal, please see our Privacy Policy .

Reviewer #1: No

Reviewer #2: No

Reviewer #3: No

**Figure resubmission:**
---

## [Decision Letter · Decision Letter 1]

9 Mar 2026

Dear Dr Djaafara,

We are pleased to inform you that your manuscript 'Dengue transmission heterogeneity across Indonesia’s archipelago: climate-driven spatiotemporal patterns and policy implications' has been provisionally accepted for publication in PLOS Neglected Tropical Diseases.

Best regards,

David Safronetz, Ph.D.

Section Editor

David Safronetz

Section Editor

Shaden Kamhawi

co-Editor-in-Chief

Paul Brindley

co-Editor-in-Chief

Reviewer's Responses to Questions

**Key Review Criteria Required for Acceptance?**

**Methods**

-Are the objectives of the study clearly articulated with a clear testable hypothesis stated?

-Is the study design appropriate to address the stated objectives?

-Is the population clearly described and appropriate for the hypothesis being tested?

-Is the sample size sufficient to ensure adequate power to address the hypothesis being tested?

-Were correct statistical analysis used to support conclusions?

-Are there concerns about ethical or regulatory requirements being met?

Reviewer #1: (No Response)

**Results**

-Does the analysis presented match the analysis plan?

-Are the results clearly and completely presented?

-Are the figures (Tables, Images) of sufficient quality for clarity?

Reviewer #1: (No Response)

**Conclusions**

-Are the conclusions supported by the data presented?

-Are the limitations of analysis clearly described?

-Do the authors discuss how these data can be helpful to advance our understanding of the topic under study?

-Is public health relevance addressed?

Reviewer #1: (No Response)

**Editorial and Data Presentation Modifications?**

Reviewer #1: (No Response)

**Summary and General Comments**

Reviewer #1: The author provided an excellent response to the question I raised.

PLOS authors have the option to publish the peer review history of their article (what does this mean? ). If published, this will include your full peer review and any attached files.

**Do you want your identity to be public for this peer review?** For information about this choice, including consent withdrawal, please see our Privacy Policy .

Reviewer #1: No

---

## [Editor Report · Acceptance letter]

Dear Dr Djaafara,

We are delighted to inform you that your manuscript, "Dengue transmission heterogeneity across Indonesia’s archipelago: climate-driven spatiotemporal patterns and policy implications," has been formally accepted for publication in PLOS Neglected Tropical Diseases.

Best regards,

Shaden Kamhawi

co-Editor-in-Chief

Paul Brindley

co-Editor-in-Chief
